# High Spatial Resolution Nitrogen Emission and Retention Maps of Three Danish Catchments Using Synchronous Measurements in Streams

Sofie G. M. van't Veen [1,2,*], Jonas Rolighed [1], Jane R. Laugesen [2], Gitte Blicher-Mathiesen [1] and Brian Kronvang [1,*]

1  Department of Ecoscience, Aarhus University, 8000 Aarhus, Denmark
2  Envidan A/S, 8600 Silkeborg, Denmark
*  Correspondence: svv@ecos.au.dk (S.G.M.v.V.); bkr@ecos.au.dk (B.K.); Tel.: +45-28257578 (S.G.M.v.V.)

**Abstract:** We investigated the utility of using synchronous measurements to create nitrogen (N) emission and retention maps of agricultural areas. Total N (TN) emissions from agricultural areas in three different Danish pilot catchments (1800–3737 ha) and within sub-catchments (100–1200 ha) were determined by a source apportionment approach. Intensive daily (main gauging stations) and fortnightly (synchronous stations) monitoring of discharge, TN, and nitrate-N ($NO_3$-N) concentrations was conducted for two years. The groundwater N retention was calculated as the difference between a model-calculated $NO_3$-N leaching from agricultural fields and the calculated agricultural N emission. The average annual N leaching and N emission in the three catchments amounted to 68, 48, and 58 kg N/ha and 6, 30, and 40 kg N/ha, respectively. The N retention in groundwater in the three catchments, calculated based on either TN or $NO_3$-N emissions, amounted to 26 and 44%, 44 and 57%, and 93 and 97%, respectively, with large variations within two of the main catchments. From this study, we conclude that synchronous measurements in streams provide a good opportunity for developing local N emission and N retention maps. However, $NO_3$-N should be used when dealing with N retention calculation at the finer resolution scale of 100–300 ha catchments.

**Keywords:** stream measurement; small-scale catchment hydrology; nitrogen emission; uncertainty; nitrogen retention; nitrogen regulation; high spatial resolution monitoring

## 1. Introduction

Eutrophication of surface waters is a major threat worldwide to obtaining a good ecological and chemical quality of water bodies [1–3]. Excess nitrogen (N) loadings from agriculture to coastal and open marine waters have been documented to cause major nutrient-limiting algal growth and blooms in Europe, New Zealand, Australia, and the US [2–6]. An example is seen in the Mississippi river basin and Chesapeake Bay in the US, where many restoration programs have been introduced to improve the environmental conditions. Nevertheless, a plethora of serious problems remains with the eutrophication of the coastal waters [3,6].

Nitrate pollution from diffuse sources in Europe causes problems in ecosystems and human health alike, due to oxygen depletion and eutrophication [7]. In a report from the European Commission, based on data from 2016 to 2019, Denmark was indicated as one of the member states having a large number of eutrophic waters, and therefore the European Commission wants stricter measures to enforce the Nitrate Directive (ND) [7]. In 2021, only 5% of Danish coastal waters were in "good ecological condition" [8,9]. Despite persistent regulation and a decrease of the total N (TN) emissions from point sources to surface waters in Denmark by more than 80% and of diffuse sources by almost 50% during the period 1990–2018, eutrophication in Danish coastal waters remains the main factor preventing the establishment of good ecological conditions [2,7,10,11].

Denmark is obliged to fulfil requirements involving monitoring and evaluating the ecological and chemical status of our water bodies as required in the ND, The Water Framework Directive (WFD) from 2000 [12], and by the Baltic Marine Environment Protection Commission (HELCOM), whose member countries have agreed a joint effort to reduce nutrient loadings to coastal waters [13].

The general regulation of N in chemical fertilizer and of N in manure in Danish agriculture has always been consistent and uniform, with the same restrictions for the entire territory [11]. This is not cost effective, and therefore, since 2015, the Danish Ministry of Environment and Food has implemented a new regulation strategy based on a more spatially-targeted N regulation [14]. This strategy includes routine mitigation measures, such as catch crops, related to the spatial variability between catchments in the natural N retention in surface and groundwater [10,11]. A new emission-based N regulation was also introduced in the Danish Agricultural Package in 2015, where farmers were allowed to carry out their own control and documentation of N emissions from fields or small catchments [14].

In the Danish targeted N regulation adopted in 2016, farmers can implement mitigation measures together, such as catch crops, constructed wetlands, restored wetlands, and afforestation [14–16]. The N reduction efforts in the targeted regulation are differentiated among the 109 coastal water catchments, and the reduction goals are defined using marine models, baseline N loadings, and targets set for good ecological quality for chlorophyll a and the depth limit of eelgrass [2]. The regulation introduced in 2015 intended to target the N reduction efforts in each coastal water catchment area (e.g., catch crops) towards catchments where the mitigation measures have the strongest effects and thus are most cost-effective.

The effectiveness of the new targeted N regulation is ensured by the Danish government, by first offering a voluntary requirement, followed by a mandatory requirement, and the expected effect is an annual reduction of approx. 3500 tons N every year in coastal catchments [17]. The targeted N regulation is based on a national N retention map for both groundwater and freshwaters, which was developed during the period 2013 to 2015 based on catchments with an average area of 1500 ha (ID15 catchments) [18]. The national N retention map indicates the proportion of the leached nitrate from the root zone expected to be reduced before it reaches the coastal waters and is estimated for the whole of Denmark, divided into the ID15 catchments [19]. However, the national N retention map is based on topographic delimitation of the ID15 catchments and does not directly show in which surface water system the mitigation measures will have the most effect because N may be transported to another ID15 catchment, due to the possibility of groundwater crossing the catchment boundaries. Therefore, the mitigation measures in one ID15 catchment may influence the N retention in another catchment [19]. This shows the importance of catchment area delimitation when estimating N retention.

In addition, although this emission-based regulation includes N retention at the ID15 catchment scale, variations in N retention and flow paths within an ID15 catchment will occur. Therefore, a better effect from mitigation measures can be obtained if it is possible to locate them in small-scale sub-catchments with a low N retention. Thus, it is interesting to investigate the possibility of quantifying N emissions and N retention on a smaller scale than the hitherto used ID15 catchments and with an acceptable uncertainty using synchronous measurements in streams.

Several studies have shown that small-scale catchment areas with a size of around 10 km$^2$ are a good platform for exploring N emissions from agriculture or forestry [15,20,21]. A study conducted in Sweden by Kyllmar et al. [22] investigated N emissions from agricultural areas in small catchments of approximately 10 km$^2$ [22]. The study showed that detailed information on agricultural management and biweekly stream measurements are appropriate as indicators of how farm practice responds to policy, regulations, and surface water quality. They also found that N emissions varied among the agricultural areas, due to soil types, climate, crops, agricultural area, and animal density [22]. The

authors [22] found that small-scale catchment monitoring led to early identification of the response of water quality to changes in agricultural practices compared to larger stream catchments. This is because the retention in streams and lakes in larger catchments is larger, and due to the higher contribution from other sources such as point sources [22]. Therefore, this observation scale can provide information on the success of regulation and the implementation of mitigation measures. Other studies have estimated groundwater retention at overall catchment level and sub-catchment level and examined variations between sub-catchments [23–25]. However, none of these studies have tried to develop high resolution N retention and N emission maps based on water sampling in streams and N leaching from agricultural fields. Both Müller-Wohlfeil et al. [24] and Wendland et al. [25] used modeling approaches to investigate catchment N leaching and N retention.

The overall aim of this study was to explore the spatial variation in N emissions and N retention at a small scale (1 to 12 km$^2$) by conducting intensive synchronous measurements of discharge and N concentrations in streams in three Danish catchments with different climate and pedology. The specific objectives of the study were (i) to investigate the spatial variation of agricultural N emission maps based on stream measurements and (ii) to explore local groundwater N retention calculated based on TN or nitrate-N (NO$_3$-N) emissions and compare this with nationally modeled N retention maps, to investigate uncertainties and differences.

## 2. Materials and Methods

### 2.1. Study Sites

Three small Danish catchments (Jegstrup, Odder, and Saltø) were chosen as study sites, to represent different conditions of geology, hydrology, soil, and agricultural practices (Figure 1). Jegstrup stream catchment (21.7 km$^2$) is located in central Jutland, Odder stream catchment (17.9 km$^2$) is located in the eastern part of Jutland, and Saltø stream catchment (37.4 km$^2$) is located in the western part of Zealand (Figure 1). Intensive data on agricultural practices, drainage conditions (e.g., where tile drains were installed in the soils), and soil types were collected for the three catchments (Table 1).

Jegstrup stream catchment is located close to the city of Viborg and is part of the Hjarbæk coastal area. The agricultural areas cover 63% of the catchment and the soils are dominated by coarse sand (86%). There are many cattle and pig farms in the catchment, and the main crop type is grain (Spring Barley) 33%. The catchment has almost no tile drains (5%). Jegstrup stream catchment is divided into three sub-catchments, varying in size between 4.5 and 11.8 km$^2$ (Table 1).

The Odder stream catchment is located close to the town of Odder. The catchment is a part of the Norsminde coastal area and 57% is agricultural area. The agricultural soils are dominated by sandy loam (55%) and clayey sandy soil (42%). The terrain is hilly moraine, and large parts of the agricultural soil in the catchment is tile-drained (76%). Many pig farms are located in the catchment, and the main crop type is grain (winter barley), of which, 62% is used as pig feed. The Odder stream catchment is divided into five sub-catchments, varying in size between 1.3 and 6.9 km$^2$ (Table 1).

Saltø stream is located close to the city of Slagelse and is a part of the Karrebæk coastal area. The terrain in the catchment is in generally flat and 73% is agricultural area, dominated by 69% sandy loam and almost fully tile-drained (97%). In the catchment, there are some pig farms, and the main crop type is grain (winter seed) 55% (Table 1). The Saltø stream catchment is divided into seven sub-catchments, varying in size between 2.6 and 9.9 km$^2$ (Table 1) [26].

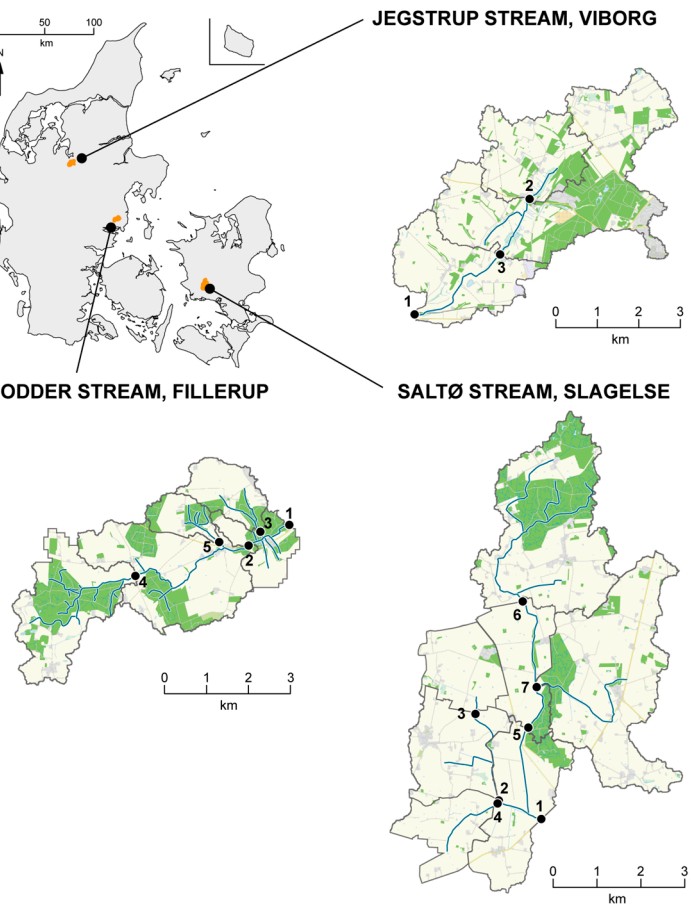

**Figure 1.** Overview map showing the location of the three study sites: Saltø stream catchment, Jegstrup stream catchment, and Odder stream catchment in Denmark. The numbers indicate the sub-catchments. Jegstrup stream catchment is divided into 3 sub-catchments, Odder stream catchment into 5 sub-catchments, and Saltø stream catchment into 7 sub-catchments.

The average annual precipitation in the three catchments in the two monitoring years October 2014 to October 2016 was in Jegstrup: 956, Odder: 900, and Saltø: 702 mm/year. For Denmark the precipitation in 2015 was 904 mm and in 2016 was 701 mm, and the annual mean temperature was 9.1 °C and 9.0 °C [27].

The delimitation of the catchments was performed using ArcGIS based on a national digital terrain model (1 × 1 m²) and supported by detailed knowledge from farmers from the tile-drained areas. Each of the three catchments was divided into several smaller sub-catchments. Jegstrup catchment was divided into three, Odder catchment into five, and Saltø catchment into seven sub-catchments (Table 1). The strategy was to divide the main catchments into smaller sub-catchments with as many headwater sub-catchments as possible. This strategy was carried out to reduce the uncertainty when calculating the N emission from each sub-catchment using intensive sampling at all headwater sub-catchments and a mass balance approach for the sub-catchments involving several monitoring stations. In Jegstrup catchment, two out of three sub-catchments are headwater sub-catchments; in Odder catchment, three out of five are headwater sub-catchments; and in Saltø catchment, four out of seven are headwater sub-catchments (Appendix C, Table A2). To simplify the principle used in the mass balance calculations, a conceptual figure showing the calculations for Odder stream sub-catchments can be found in Appendix C, Figure A4.

**Table 1.** Overview of the sub-catchments in the three study areas of Jegstrup, Odder, and Saltø.

| Catchment ID | Area (km²) | Precipitation, 25 October 2014–24 October 2015 (mm) | Precipitation, 25 October 2015–24 October 2016 (mm) | Main Agricultural Soil Type (Danish JB-Soil Classification System) | Agricultural Areas (%) * | Drain Areas (% of the Agricultural Area) ** | N Application (Agricultural Land). Commercial and Manure (kg ha) | Main Crop Type (2015) |
|---|---|---|---|---|---|---|---|---|
| 1 | 5.4 | 955 | 965 | | 77 | 7 | 162 | Winter Rye |
| 2 | 4.5 | 939 | 966 | | 68 | 2 | 167 | Spring Barley |
| 3 | 11.8 | 944 | 966 | | 54 | 4 | 171 | Spring Barley |
| **Total catchment Jegstrup stream** | **21.7** | **946** | **966** | **coarse sandy soil 86%** | **63** | **5** | **167** | **Spring Barley (33%)** |
| 1 | 2 | 830 | 897 | | 59 | 77 | 128 | Winter Wheat |
| 2 | 6.9 | 870 | 951 | | 72 | 76 | 159 | Winter Wheat |
| 3 | 2.2 | 869 | 950 | | 71 | 93 | 183 | Winter Barley |
| 4 | 5.4 | 869 | 947 | | 55 | 66 | 134 | Winter Wheat |
| 5 | 1.3 | 870 | 951 | | 54 | 89 | 166 | Spring Barley |
| **Total catchment Odder stream** | **17.9** | **865** | **944** | **sandy loam 55%, clayey sandy soil 42%** | **57** | **76** | **160** | **Winter Wheat (62%)** |
| 1 | 4.9 | 683 | 719 | | 79 | 98 | 194 | Winter Wheat |
| 2 | 4 | 683 | 719 | | 87 | 97 | 167 | Winter Wheat |
| 3 | 3 | 683 | 719 | | 88 | 99 | 180 | Winter Wheat |
| 4 | 2.7 | 683 | 719 | | 94 | 99 | 193 | Winter Wheat |
| 5 | 3.5 | 683 | 719 | | 83 | 98 | 183 | Winter Wheat |
| 6 | 9.5 | 683 | 719 | | 55 | 96 | 162 | Winter Wheat |
| 7 | 9.9 | 684 | 725 | | 80 | 98 | 155 | Winter Wheat |
| **Total catchment Saltø stream** | **37.4** | **683** | **721** | **sandy loam 69%** | **77** | **97** | **173** | **Winter Wheat (55%)** |

Note: * from [28]. ** from [29].

*2.2. Discharge Measurement and Water Sampling in the Streams*

In all three catchments, a main gauging station was installed at the outlet. Within the catchments, monitoring stations were installed in the streams at the outlet of each sub-catchment, and measurements of discharge and nutrient concentrations were conducted during two hydrological years—year 1: October 2014 to September 2015, and year 2: October 2015 to September 2016.

The water level (H) was continuously monitored (10 min) at the three main gauging stations using a pressure transducer (a MagdeTech Level 2000 was installed in Saltø stream and Jegstrup stream, and a TD-Diver van Essen Instrument was installed in Odder stream). Instantaneous measurements of the discharge (Q) were conducted at weekly intervals in 2014 and 2015 and fortnightly in 2016 using an Ott MF Pro instrument and an ADCP Streampro from Teledyne Instruments. The daily mean discharge was calculated using an established stage (H)-discharge (Q)-relationship (Q/H) for each gauging station, with corrections for weed growth in the summer period [30]. In each sub-catchment, synchronous monitoring stations were established and discharges were measured fortnightly, and the linear relationship to the discharge calculated on the same day at the main gauging station was established (Q-Q relationships). The established Q-Q relationships generally have a high explanation value, where the majority have an $R^2$ > 0.9 (Appendix A, Table A1). These Q-Q relationships were used to calculate the daily mean discharge at the synchronous stations for the two monitoring years [30].

Grab sampling was conducted every week at the synchronous monitoring stations during the first monitoring year and every third week in the second monitoring year. All water samples were kept cold (at a temperature of 5 °C) and dark and brought to the laboratory within 24 h. The sampling strategy for the synchronous monitoring stations was to measure the discharge and nutrient concentrations intensively in as many headwater sub-catchments as possible, to reduce the uncertainty when estimating the N transport for the individual sub-catchments. A study by Kronvang and Bruhn [31] found that using a regular fortnightly monitoring strategy resulted in high accuracy calculation of the annual TN transport in small lowland streams (RMSE at 1.1 to 4.9%).

At the main gauging stations, water samples were collected by an automatic sampler (ISCO 6700 FR in Saltø stream and Jegstrup stream) equipped with a refrigerator, to keep the water samples at a temperature of 5 °C and in the dark. In the Odder stream, an ISCO 3700 was used. A sample was taken every three hours and mixed into daily composite samples. All the collected water samples were analyzed for TN, $NO_3$-N, and nitrite N ($NO_2$-N) in the laboratory at Aarhus University following the same procedures as in [32]. TN was analyzed following DS/EN 12260 (2003) [33] using a TOC-L analyzer with a TNM-L module (Shimadzu, Kyoto, Japan) at a temperature of 720 °C. $NO_3$-N samples were filtered in the laboratory using Whatman GF/C filters and analyzed according to DS/EN ISO 10304 (2009) [34] using ion chromatography (Dionex ICS-1500 IC-system, Thermo Fisher Scientific, Roskilde, Denmark). An anion Micro Membrane Suppressor (AMMS III 4 mm) was used as the basis eluent, and the method used a separator column (IonPac AS22), a guard column (IonPac AG22), and an eluent, which was a mixture of 4.5 mM $Na_2CO_3$ and 1.4 mM $NaHCO_3$. All samples for $NO_3$-N analyses were filtered in the system according to (SNY2225, Frisenette ApS, Knebel, Denmark) through a filter consisting of a double-layered 0.22 μm glass fiber filter.

The annual runoff (mm) was calculated for all the sub-catchments based on the annual discharge and the sub-catchment areas, together with the mass balance principle (Appendix C, Table A2, and Figure A4). A baseflow index (BFI) was made from the calculated daily mean discharges at the three main gauging stations according to Gustard et al. [35].

*2.3. $NO_3$-N Leaching from Agricultural Areas*

$NO_3$-N leaching from the root zone was calculated at field level using the empirical Nitrogen Leaching Estimator model N-LES4 for the two hydrological years 2014/15 and 2015/16 and subsequently aggregated to field block and catchment level [36,37]. The N-LES model is regularly updated [38], and the N-LES4 model was used in previous studies to calculate the N leaching from agricultural fields [37,39,40].

N-LES4 calculates the $NO_3$-N leaching from the root zone 1 m below the soil surface based on the amount of water that infiltrates through the root zone. Input parameters cover the N application and crops in the leaching year, the average N application during the preceding five years, and information about soil type and percolation during the last two years, together with the crops in the previous year [36]. The N-LES4 model was based on 1467 measurements of annual $NO_3$-N leaching, which included 1058 measurements from different Danish research stations, together with 409 measurements from farm field sites in five monitoring catchments in Denmark [37].

The N-LES4 model has not been validated in relation to uncertainty using independent data or any kind of cross-validation [36]. However, the previous version of N-LES3 was cross-validated by excluding a subset of the measured input data followed by re-estimation of the leaching function parameters. The results showed that the uncertainty between the estimated and measured data was 10 to 30% of the predicted $NO_3$-N leaching using more than one field or years [37,41]. Furthermore, the validation showed that the N-LES parameters were generally stable, i.e., not sensitive to differences in the empirical data input [37,41]. Combined with the uncertainties of the model input data, we can only assume that the total uncertainty of the predicted $NO_3$-N leaching must be as least as high as stated in the N-LES3 documentation [37,41].

Model inputs for agricultural practices, including fertilizer application and crop coverage at field-block level, were based on single payment registers and fertilizer accounts, which are compulsorily submitted to the Danish authorities by farmers every year. The two datasets were linked by means of a common farm identity number or a common farm address, and the reported amounts of fertilizer and manure from the individual accounts were distributed for the fields of each farm according to the crop N standards.

The grass-ley area was derived from the area with rotation grass, assuming a conversion rate of three years, provided that there was enough space in the crop rotation. The catch crop area was based on data from the fertilizer accounts. Standard values for nitrogen fixation were used for each crop.

Water percolation from the root zone was calculated using the Daisy model [42] applying daily climate data from a 10 km grid net covering Denmark. The daily climate data were provided by the Danish Meteorological Institute.

### 2.4. TN and NO$_3$-N Transport and Emission

The daily transport of TN and NO$_3$-N was calculated based on the product of mean daily discharge, and either mean daily concentrations or linearly interpolated concentrations were measured according to [5,12]. The annual transport of TN and NO$_3$-N for each sub-catchment was bias corrected with deviations between the grab sampling and intensive sampling derived for the three main gauging stations at the catchment outlets (Appendix B, Figures A1–A3) and Section 2.6.

Flow-weighted TN concentrations were calculated for all monitoring stations by dividing the calculated annual TN transport by the annual discharge.

N emissions from the agricultural areas in the sub-catchments (diffuse sources) were calculated using a source apportionment of both the annual TN and NO$_3$-N transport from the catchments [28,30]. As input to the source apportionment, all emissions of N from known point sources in the catchments (wastewater treatment plants, scattered dwellings, and stormwater outlets) were collected from the Danish national wastewater database provided by the Danish Environmental Protection Agency [43,44]. Data on N retention in surface waters (streams, wetlands, and lakes) were extracted from the Danish National Nitrogen Model [18,45]. Data on the atmospheric deposition of N on surface waters in the catchments were obtained from the National Atmospheric Deposition Model [46,47]. Source apportionment of the annual TN and NO$_3$-N transport from the catchments was carried out using the following mass balance equation [30]:

$$T_0 = L_E - L_E R_L + P_E - P_E R_P + B_E - B_E R_B + A_E - A_E R_A, \qquad (1)$$

where $T_0$ = N export from the catchment (the TN and NO$_3$-N transport in the stream), $L_E$ = emissions from the agricultural areas in the catchment to surface water, $P_E$ = emissions of N to surface water from all point sources in the catchment area (wastewater treatment plants, scattered dwellings, and stormwater outlets), $B_E$ = N emissions from natural areas in the catchment (also referred to as natural background emission of N), $A_E$ = atmospheric deposition directly on open water surfaces in the catchment area, and $R_x$ = retention of N in surface water in the catchment. This means that the measured TN or NO$_3$-N transport ($T_0$) is the sum of all sources in the catchment, minus the retention that each source has been affected by on the way through the freshwater system in the catchment. The total amount of N removed by the retention of each source is referred to as $R_0$. The emission of N from agricultural areas can therefore be estimated using the following equation [28,30]:

$$L_E = T_0/(1 - R_0) - P_E - B_E - A_E, \qquad (2)$$

### 2.5. N Retention in Groundwater

The N retention in groundwater in the sub-catchments in the two monitoring years was calculated as the difference between the modeled average N leaching (NO$_3$-N) from the root zone (upper 1 m) in agricultural land and the N emission (TN and NO$_3$-N) from

agricultural areas to streams for all sub-catchments [30]. We assumed that the $NO_3$-N leaching from the root zone at 1 m depth was leached into the saturated groundwater zone in the sub-catchments and by that towards the groundwater (Section 2.3). This assumption was based on the Danish national map showing the depth from the surface to the groundwater in 100 m grid size, based on measurement of groundwater from 1990 to 2020 [48]. The national map showed that the depth from the surface to groundwater in Saltø catchment mainly was 0.5 to 1 m and for smaller areas 0 to 0.5 m. In the Odder catchment, the depth from the surface to groundwater was mainly 0.5 to 1 m and for smaller areas 1 to 2 m. In the Jegstrup catchment, there was more variations, from 0.5 to 1 m mainly in the northern part of the catchment, to >10 m in the southern part of the catchment [48]. The mean annual N retention in groundwater was calculated based on the N retention in each of the two monitoring years. The overall groundwater N retention for each total catchment was calculated for both TN and $NO_3$-N based on the $NO_3$-N leaching for all sub-catchments and the N emission measured at the main gauging station in each catchment. The groundwater N retention in this study is referred to as the removal of N due to denitrification, together with what is still "held back" in the catchment due to the transport time from the field to the stream edge.

### 2.6. Statistical Methods for N Transport Calculation

A Monte Carlo method was used to investigate the uncertainty of using grab samples for measuring $NO_3$-N and TN concentrations in the streams to calculate the annual N transport. The annual N transport calculated from the automatically sampled mean daily composite water samples at the three main stations was defined as the "true" transport. Thereafter, the differences obtained for the annual N transport were evaluated by simulating four different sampling strategies for TN only, as $NO_3$-N was the major constituent of TN (64% in Jegstrup stream, 79% in Odder stream, and 80% in Saltø stream): (1) weekly; (2) fortnightly; (3) fortnightly in the winter period and monthly in the summer period (18 annual samples); (4) monthly using the Monte Carlo method based on the daily dataset of N concentrations at the three main gauging stations. For each simulated sampling strategy, the linear interpolation method was used to estimate daily N concentrations. The simulations of the annual N transport based on the different sampling strategies were then compared to the "true" annual N transport. The analysis showed a deviation from the "true" annual N transport, and therefore the annual N transport needed to be bias corrected with the deviations found for the three main stations (−2.6 to 0.9%) [28,49]. The bias arose because grab samples were taken at the synchronous monitoring stations using a low-frequency measuring strategy for analysis of TN and $NO_3$-N, as described in Section 2.2. This means that all the calculated annual N transports were originally underestimated. The annual TN and $NO_3$-N transports for the synchronous stations in Saltø catchment were therefore, before source apportionment of the annual N transport, bias (deviation) corrected by −1.1% for year 1, due to weekly grab samples, and for year 2 by −4.1%, due to grab samples being taken every three weeks. For the synchronous stations in Odder catchment, the annual transport was bias corrected by −2.6% for year 1 and −6.2% for year 2. For the synchronous stations in Jegstrup catchment, the annual transports were bias corrected by −0.3% for year 1 and 0.6% for year 2 [31,49].

### 3. Results

### 3.1. $NO_3$-N Leaching from Agricultural Areas

The average $NO_3$-N leaching from agricultural areas for the two monitoring years amounted to 48 kg N/ha in Saltø catchment, 58 kg N/ha in Odder catchment, and 69 kg N/ha in Jegstrup catchment (Figure 2). The $NO_3$-N leaching in Jegstrup catchment was higher than in the other two catchments, which might be attributable to the dominance of sandy soils (86%) and higher use of animal manure, while the soils in Odder and Saltø catchments contain more clayey soil. The difference in soil type between the catchments also explains why the percentage of drained area is lower in Jegstrup catchment than in the

other two catchments (Table 1). The highest leaching in the Jegstrup catchment was found in the upstream sub-catchment 2. In the Odder sub-catchments, the $NO_3$-N leaching varied from 50 to 63 kg N/ha, and the lowest leaching was seen in the downstream sub-catchments 1, 3, and 5. In the Saltø sub-catchments, the $NO_3$-N leaching varied more between the sub-catchments than in the two other catchments. Thus, the leaching varied from 34 to 59 kg N/ha, and the upstream sub-catchment 6 had the highest leaching (Figure 2).

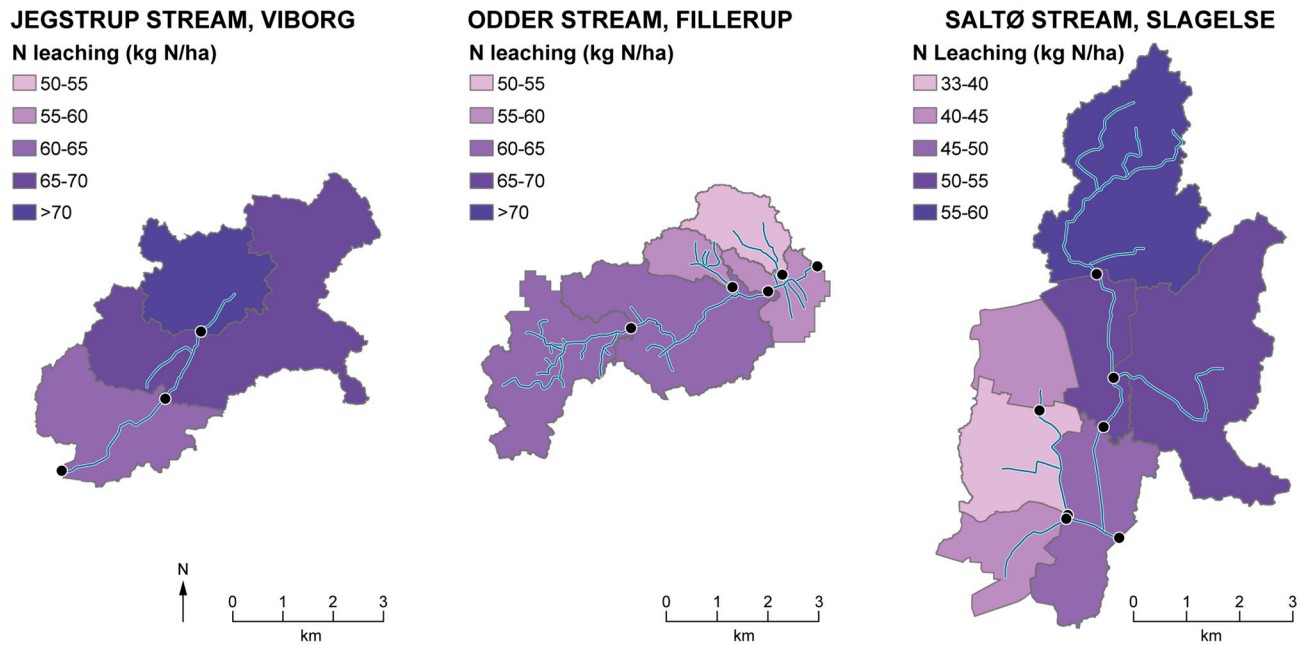

**Figure 2.** Average nitrate ($NO_3$-N) leaching from agricultural areas from 2015 to 2016 in the three catchments Jegstrup, Odder, and Saltø.

*3.2. Catchment Hydrology and N Concentrations*

The Jegstrup stream had a BFI index of 0.81, which indicated that the stream hydrology was primarily dominated by groundwater. The discharge was almost stable during the year, with minimal differences between the maximum and minimum daily average discharge (Figure 3B). Additionally, the measured daily average TN and $NO_3$-N concentrations were in general low and stable during the monitored years, with only short peak periods caused by supply of N to the stream in periods with high discharge (Figure 3A). In the Jegstrup catchment, the spatial distribution of the runoff varied between sub-catchments, and was highest in the downstream sub-catchment 1 (Figure 4). This is probably due to the influence of urban areas in the eastern part of the catchment and the location of groundwater abstraction sites in the north.

The Odder stream had a BFI index of 0.56, and both quick flow (drain water) and base flow (groundwater) were thus important hydrological pathways to the stream. The discharge varied during the year, with the highest discharge occurring in autumn and winter (Figure 3B). Odder stream experienced large variations throughout the year in daily average TN and $NO_3$-N concentrations, with the concentrations being highest in winter and lowest in summer (Figure 3A). The highest runoff was recorded in Odder catchment (Figure 4). Moreover, the spatial variation of runoff in the Odder stream was large, being highest in the downstream sub-catchments (Figure 4).

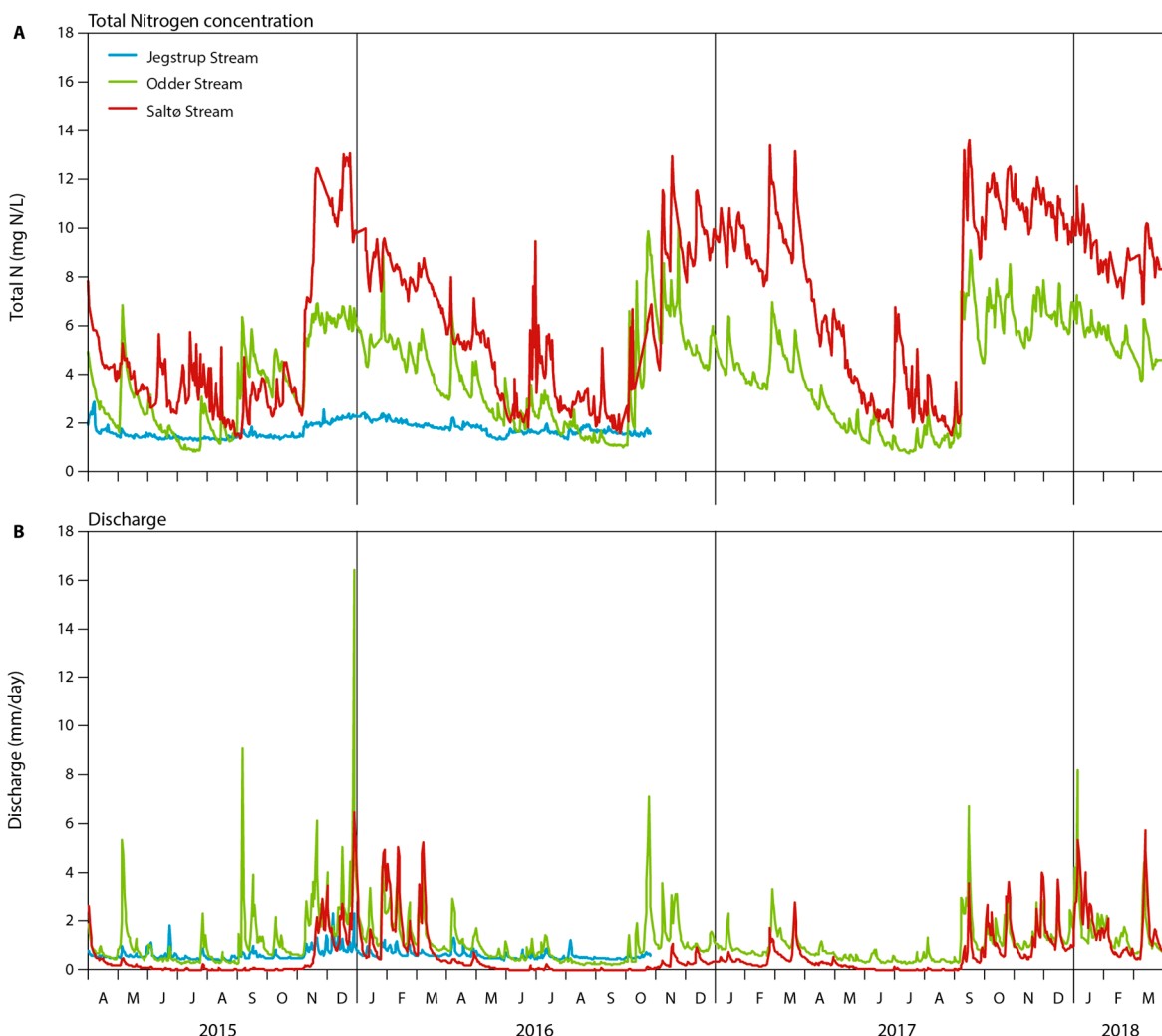

**Figure 3.** (**A**) Variations in TN concentration from 2015 to 2016 measured in the Jegstrup stream, Odder stream, and Saltø stream. (**B**) Variation in the measured discharge from 2015 to 2016 in the Jegstrup stream, Odder stream, and Saltø stream. Modified from [46].

The Saltø stream had a BFI index of 0.40, which indicated that fast-responding drain water dominated the stream hydrology, while the interaction with groundwater was limited. The discharge was also very low in the summer period, due to the limited supply of groundwater to the stream. The daily average discharge also showed pronounced variations during the year, with the highest discharge in winter and autumn (Figure 3B). The measured daily average TN and NO$_3$-N concentrations also varied during the year (Figure 3A). The large and nearly complete systematic tile drainage of fields in the catchments indicates that most of the N leaching from the agricultural fields led directly to the Saltø stream through tile drains, thus bypassing the redox zone in the groundwater and the potential reduction of NO$_3$-N below this. The runoff in Saltø catchment ranged from 191 to 331 mm, with the maximum runoff in the downstream sub-catchment 1. The increase in runoff can probably also be ascribed to drainage of a larger area of low-lying organic-rich soil via an established pumping station (Figure 4). The measurement of discharge and TN at the synchronous stations in all sub-catchments in the Jegstrup, Odder, and Saltø catchments are found in Appendix B (Figure A1A,B, Figure A2A,B, Figure A3A,B).

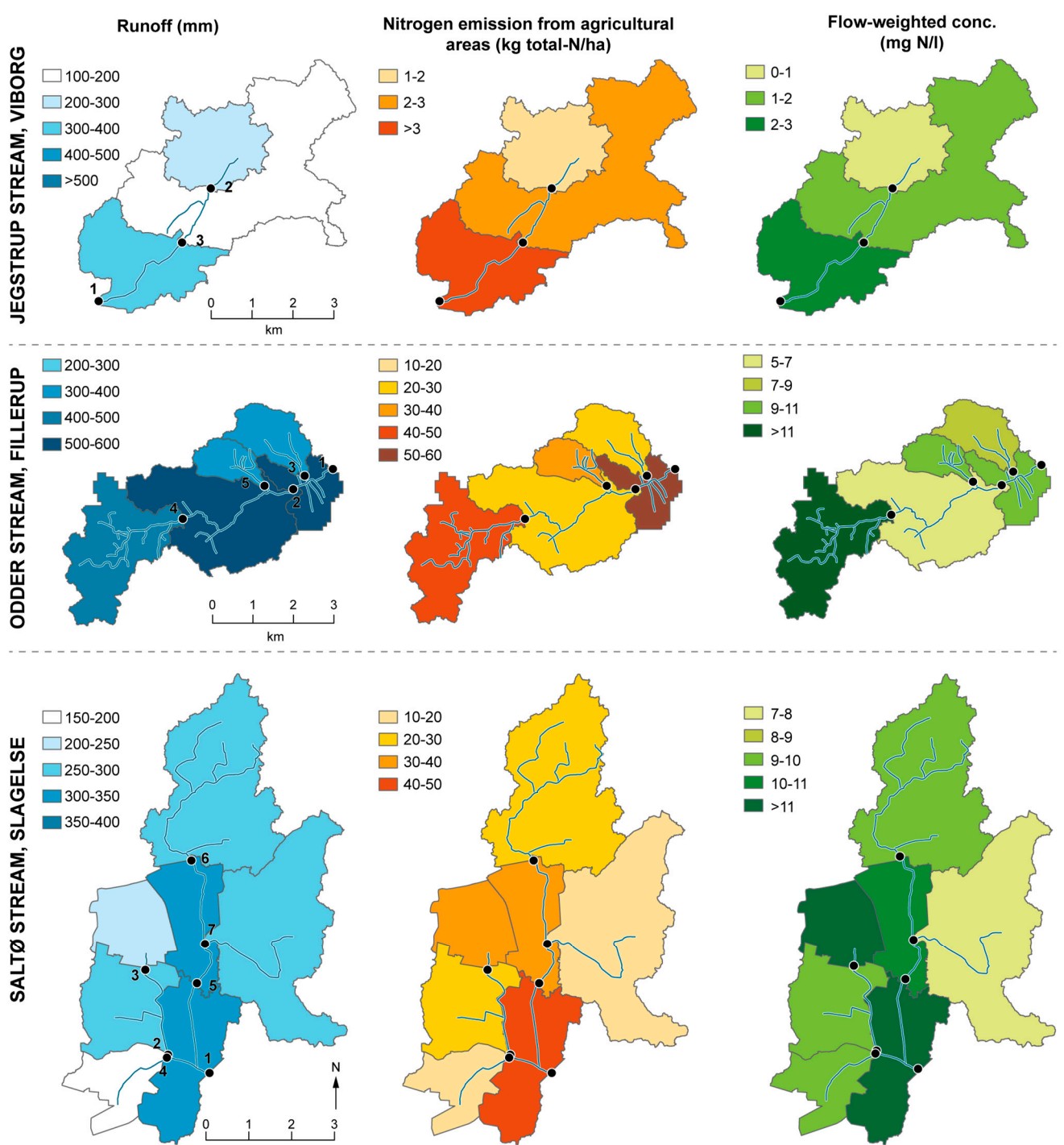

**Figure 4.** Average calculations of runoff, emission of total N (TN) from agricultural areas to stream edge, estimated by source apportionment and flow-weighted concentration of TN from agricultural areas after bias correction of the TN transport, in the two monitoring years in the sub-catchments of the three main catchments of Jegstrup (3 sub-catchments), Odder (5 sub-catchments), and Saltø (7 sub-catchments). Modified from [28].

### 3.3. N Emission from Agricultural Areas

A local average annual TN emission map for diffuse TN losses from agricultural areas is seen in Figure 4. The average annual TN emission was 4.4 kg N/ha in the Jegstrup catchment, 40 kg N/ha in Odder catchment, and 30 kg N/ha in Saltø catchment. The map shows the TN emissions to the stream edge in each sub-catchment. The TN emission

from the agricultural areas to the surface water in the Jegstrup stream was relatively small in all sub-catchments (in two of the three sub-catchments <3 kg N/ha), as were the flow-weighted concentrations of TN (Figure 4). In the Odder catchment, the TN emission from the agricultural areas to surface water was relatively large, as was the runoff, with a large spatial variation between sub-catchments, ranging from 28 to 59 kg N/ha (Figure 4). Sub-catchment 1 in the Odder catchment had a relatively high proportion of organic N (26% in year 1 and 50% in year 2), together with a high annual runoff, indicating high contributions from a deeper reduced aquifer. The proportion of organic N in the remaining sub-catchments were on average 14% in year 1 and 19% in year 2. However, the low $NO_3$-N amount in sub-catchment 1 compared to the other sub-catchments may also indicate a high removal by denitrification in reduced groundwater sediments in this sub-catchment. The flow-weighted TN concentrations in the incoming water from the agricultural areas also show a relatively large variation between the sub-catchments, from 5 to 11.4 mg N/L (Figure 4).

In the Saltø catchment, the TN emission from agricultural areas to the stream edge varied from 17 to 50 kg N/ha (Figure 4). The flow-weighted TN concentrations varied by approx. a factor of two within the catchment area (7 to 17 mg N/l), with the highest concentrations occurring in the western sub-catchment 3 and the southern sub-catchment 1 (Figure 4). Sub-catchment 1 in the Saltø stream catchment includes an area with farmed organic low-lying soil that is artificially drained, by pumping out excess water from the sub-catchment at regular intervals. This sub-catchment had a relatively high proportion of organic N (25% in year 1 and 30% in year 2) compared to the other sub-catchments, where the proportion of organic N on average was 17% in year 1 and 20% in year 2.

### 3.4. Groundwater N Retention Mapping

The calculated mean groundwater N retention, especially in the Odder stream and Saltø stream, differed substantially depending on the use of either TN or $NO_3$-N for the resulting emission to surface waters (Figure 5).

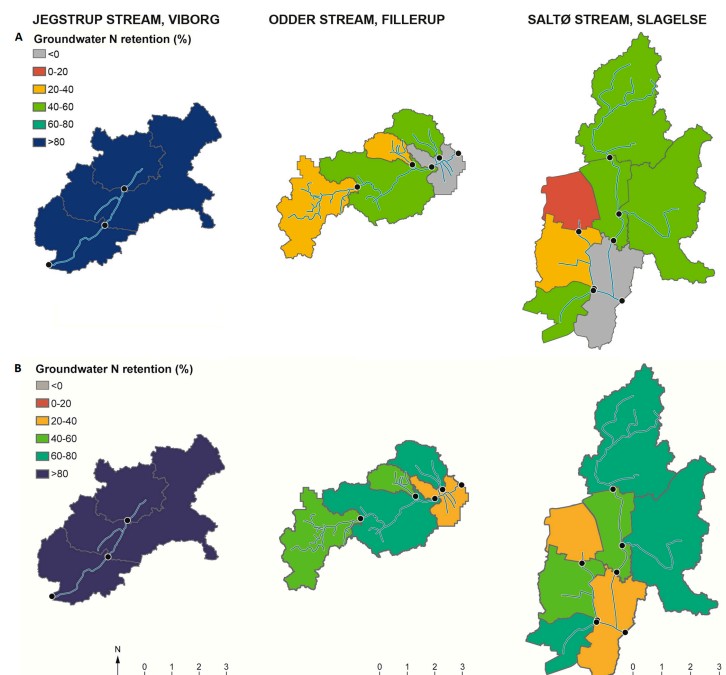

**Figure 5.** N retention in groundwater in the three catchments of Jegstrup, Odder, and Saltø. (**A**) Average TN retention in groundwater from 2015 to 2016 in the sub-catchments. The retention was calculated based on bias-corrected TN emissions from agricultural areas. (**B**) Average nitrate-N ($NO_3$-N) retention in groundwater from 2015 to 2016 in the sub-catchments. The retention was calculated based on bias-corrected $NO_3$-N emissions from agricultural areas.

The results of the calculated N retention in the groundwater in the sub-catchments of the Jegstrup catchment showed that the N retention in groundwater was larger than 80% in all sub-catchments, independently of using TN or $NO_3$-N (Figure 5). In the sub-catchments of the Odder stream catchment, the calculated N retention differed notably for the synchronous measurements, varying from $-18$ to 53% when using TN (Figure 5A) and 38 to 62% when using $NO_3$-N (Figure 5B). In the Saltø stream sub-catchments, the N retention varied from $-1$ to 60% when based on TN, and 28 to 69% when using $NO_3$-N in the calculations (Figure 5). The groundwater N retention based on either TN or $NO_3$-N in the three catchments amounted to 93 and 97% in the Jegstrup catchment, 26 and 44% in Odder catchment, and 44 and 57% in Saltø catchment.

## 4. Discussion

### 4.1. Groundwater N Retention

This study focuses on how far we can get by only using a simple semi-distributed method based on synchronous stream measurements at the outlet from sub-catchments, paired with $NO_3$-N leaching within the sub-catchments, to determine groundwater N retention. Other studies have mainly used a combination of $NO_3$-N leaching combined with the diffuse TN losses when calculating groundwater N retention [18,24,45].

The N retention in groundwater in this study was calculated by comparing the leaching of $NO_3$-N from agricultural areas with the emissions of either TN or $NO_3$-N from agricultural areas into the stream, to showcase the significant impact that organic N might have on the outcome of such a simple N balance method when used with these small-scale catchments.

Two of the sub-catchments yielded a negative N retention in groundwater when the N balance was conducted for TN (Figure 5A). This was not real and may have been because organic N (TN minus $NO_3$-N) in these two sub-catchments constituted a large part of the TN (sub-catchment 1 in Odder catchment 26% in year 1 and 50% in year 2; and sub-catchment 1 in Saltø catchment 25% in year 1 and 30% in year 2) together with a high runoff and N emissions.

An analysis of the proportion of organic N in water samples taken in the period 1990 to 2016 from 44 streams in Danish agricultural catchments (area size of 0.6 to 65 km$^2$ with average of 71% agriculture) showed that organic N on average amounted to 9 to 15% of the TN concentrations [50]. Another analysis of water samples taken from 153 regular Danish stream stations in the periods 1990 to 2008 and 2016 to 2022 showed that organic N on average amounted to 2 to 10% of the TN concentrations [51]. This means that the organic N proportion in the Saltø sub-catchment 1 and Odder sub-catchment 1 was high, not only compared to the other sub-catchments, but also compared to analyses from Danish streams.

In addition, the negative TN retention may also be because of incoming regional groundwater in the sub-catchment coming from outside the defined sub-catchment boundaries. The BFI calculations showed that groundwater constituted 81% of the discharge in the Jegstrup stream, 56% in Odder stream, and 40% in the Saltø catchment (Section 3.2), which may have contained organic N. This highlights the influence of uncertainties associated with the catchment delimitation in small-scale catchments when groundwater flow is not taken into account. The accuracy of the catchment area delimitation and the number of measurement years included in the calculation are both factors that are essential for the final accuracy of local groundwater N retention determined from local stream measurements compared to $NO_3$-N leaching from the root zone in agricultural areas [52]. However, the uncertainty related to the estimated $NO_3$-N leaching using the N-LES4 model also affects the final accuracy of the local groundwater N retention. The uncertainty of the N-LES4 model was estimated as a minimum of 10 to 30% of the predicted $NO_3$-N leaching (Section 2.3 [37,41], which is therefore also applicable to the groundwater N retention maps utilized in this study. Moreover, the uncertainty related to the monitoring strategy and frequency of sampling used for the N transport calculations was minimal, due to the bias correction of the N transport measured at the synchronous stations as a result from the

Monte Carlo analysis of the calculated N transport in the Jegstrup, Odder, and Saltø streams (Section 2.6).

In this study, the groundwater N retention for the sub-catchments was calculated based on measurements from two years (Figure 5). An analysis of the importance of the number of monitoring years shows that the mean deviation (bias) from the "true retention" is $+/-10$ percentage points using two years of stream measurements [52]. In [52] they showed that using 10 years of monitoring data results in a mean deviation of 4% from the "true" calculated N retention based on a 20 year period. This is due to annual variations in precipitation/discharge and delays in the transport of oxic and nitrate-containing groundwater from soil to streams. Thus, it is recommended to include more monitoring years to decrease the uncertainty when calculating local groundwater retention and to include groundwater flow across the catchment boundaries by, e.g., use of groundwater models. This can help make sure the catchment boundaries are defined with high accuracy.

The N retention in groundwater varied greatly among the sub-catchments, both in the Odder catchment and Saltø catchment, independently of whether the N retention was calculated based on TN or $NO_3$-N balances (Figure 5). The same catchments showed high variations in the measured TN stream concentrations during the year (Figure 3A). On the contrary, the N retention in groundwater was high, with little variation in the Jegstrup sub-catchments (Figure 5). The use of $NO_3$-N in the calculation of N retention in groundwater yielded, as expected, higher values and helped us to avoid the inaccurate negative groundwater N retention that was calculated for two sub-catchments when using TN in the N balance approach.

Therefore, we argue for using $NO_3$-N emissions instead of TN emissions to calculate N retention in groundwater when dealing with N balances on such a small scale ($<10$ km$^2$). At this scale, the transformation of, e.g., $NO_3$-N in surface water due to biological uptake and denitrification can be assumed to have a minimal impact on the N dynamics in surface waters. The conditions for denitrification are influenced by general conditions such as the amount of nitrate load, temperature, discharge, and other ecological conditions, e.g., oxygen concentration and the availability of organic substrate for bacteria [53]. Analysis of the denitrification in streams shows that the annual N removal in streams on average is below 2% of the incoming agricultural N emission to the streams and does not happen during winter when the temperature is low [53]. The highest denitrification was found in streams with mud and weeds, and the lowest rates were in sandy catchment streams without weeds [53].

Calculating the groundwater N retention using $NO_3$-N yielded a higher N retention for all three catchments, which ranged from 93 to 97% in the Jegstrup catchment, 26 to 44% in the Odder catchment, and 44 to 57% in the Saltø catchment.

A study from New Zealand also considered local groundwater N retention, based on inorganic N load calculations, and found variations between sub-catchments in the Rangitikei catchment, from 67% to 94% [23]. This variation in groundwater N retention using $NO_3$-N balances is lower than the variations found in the sub-basins of the Odder and Saltø catchments but comparable to those found between the Jegstrup sub-catchments (97 to 100%). The Rangitikei catchment groundwater N retention was, on average, 84% [23]. This is comparable with our average groundwater N retention based on $NO_3$-N of 97% in the Jegstrup catchment and higher than the average groundwater N retention in the Odder (44%) and Saltø (57%) catchments.

A case study from Mariager Fjord catchment in Denmark [24], located 55 km from Jegstrup catchment, found an average annual subsurface N retention amounting to 51% for the entire 572 km$^2$ catchment. The N retention was based on calculated $NO_3$-N leaching and measured TN stream concentrations in the sub-catchments. In addition, large variations in N retention between 10 sub-catchments in the Mariager Fjord catchment were recorded, varying from 8% to 62% [24]. The estimated N retention in that study may have been underestimated, due to the use of TN emissions in the N retention calculation; however,

their analysis was conducted at a larger catchment scale and all of their sub-catchments were larger than 10 km$^2$, which means that the transformation of, e.g., NO$_3$-N in surface water could have an impact on the N dynamics and thus on the calculated N retention in these sub-catchments [53]. In addition, a study by Wendland et al. [25] revealed that the overall groundwater retention in two catchments (one in Denmark and one in Germany) was higher than 90%, also exhibiting extensive variations between the sub-catchments [25]. Their study was, however, conducted using the SOIL-N/WEKU model [25] and was based on variations in the amount of denitrification and groundwater residence time within the two catchments and combined the soil N leaching with groundwater residence time and denitrification [25].

A comparison of our calculated groundwater N retention with the groundwater N retention from the official Danish retention map based on ID15 catchments and the national DK-model [18,45,54] shows large differences for two of the catchments (Odder and Saltø) and nearly the same values for one catchment (Jegstrup). In the Jegstrup catchment, the national N retention map shows a groundwater N retention varying from 77 to 80% in the three sub-catchments, while our estimated groundwater N retention was above 80% for all sub-catchments and 93% for the total catchment in the case of TN and 97% for NO$_3$-N (Figure 5). In the Odder catchment, the groundwater N retention varies from 71 to 78% according to the national N retention map, whereas our estimated groundwater N retention for the total catchment was 26% in the case of TN and 44% for NO$_3$-N. The calculated N retention varied even more between the sub-catchments (−18 to 53% for TN and 38 to 62% for NO$_3$-N). In the Saltø catchment, the national retention map shows that the groundwater N retention ranges between 38 and 42%, whereas our estimated N retention for the total catchment was 44% for TN and 57% for NO$_3$-N, again with pronounced variation at the sub-catchment scale (−1 to 60% for TN and 28 to 69% for NO$_3$-N) (Figure 5).

The uncertainty of the national calculated retention values for the ID15 catchments was, on average, +/−19 percentage points [18]. However, taking into account this uncertainty, together with the uncertainty related to the number of monitoring years of +/−10 percentage points [52], the calculated groundwater N retention is still different compared to the national retention map in both the Odder and Saltø catchments.

These findings indicate that local stream measurements can be used to validate groundwater retention models and contribute to gaining a higher precision when creating high resolution groundwater N retention maps. This is of great importance when shifting towards a more targeted N regulation. Additionally, this finding is in line with Singh et al. [23], who concluded that N retention maps can be useful planning tools to reduce N emissions from agricultural areas in the future [23].

A study by Hashemi et al. [11] showed that a combination of targeted mitigation measures and a change of agricultural management can reduce N emission to aquatic ecosystems, considering the spatial constraints for implementing such targeted measures. In addition, they found that a fine spatial resolution of the groundwater reduction map is necessary to achieve efficiency in the targeting of measures [11].

However, groundwater retention is affected by both hydrological and biogeochemical processes, such as water transport through the soil, geological layers, topography, and denitrification [55]. Therefore, other approaches, such as numerical modeling and conservative tracer experiences [55], could be included to investigate N transformation and N transport in more detail in sub-catchments and supplemented with geochemical data to evaluate temporal and spatial differences [55].

*4.2. N Emission and Danish Water Quality Targets*

In Denmark, there are presently no official requirements or targets for N emissions into the streams in a catchment at local geographical scale under the WFD river basin management plans (RBMP). The national targets are thus set for entire coastal water catchments [18].

Comparison of the estimated N emissions based on the N emission map with the targets for the coastal waters for the three catchments according to the Danish RBMPs 2015–2021 (Table 2) shows that the targets for Jegstrup catchment have already been achieved. Contrarily, a large reduction of normalized N emissions from agricultural areas is needed for both Odder catchment (approx. 70%) and Saltø catchment (approx. 50%) (Table 2). Normalization means that the N emission is corrected for year to year differences in the discharge in the streams, and thus the climate year to year differences are minimized.

**Table 2.** Target fulfillment for 2027 in the three final recipients of the three study streams based on the Danish river basin management plans 2015–2021 [9,56,57]. Calculated normalized N emission from the agricultural areas minus the surface water retention for the reference period 2000–2015 is also given. Modified from [28].

| Catchment | Coastal Area (Final Recipient) | Maximum Average Emission from Agricultural Areas at Target Emission 2027 According to River Basin Management Plans 2015–2021 (kg N/ha) | Normalized N Emissions from Agricultural Areas Minus the Retention (kg N/ha) |
|---|---|---|---|
| Jegstrup stream | Hjarbæk | 9.7 | 3.7 |
| Odder stream | Norsminde | 6.6 | 19.4 |
| Saltø stream | Karrebæk | 10.7 | 21.9 |

A comparison of the N emissions for each sub-catchment with the targets shows that all sub-catchments in the Jegstrup catchment achieved both the maximum average emission target for 2027 of 9.7 kg N/ha and the normalized N emission target of 3.7 kg N/ha (Figure 4; Table 2). In the Odder and Saltø catchments, none of the sub-catchments met the emission targets of the RBMP (Figure 4; Table 2). Therefore, mitigation measures should be implemented in all sub-catchments to reduce N emissions. Thus, our N emission maps could be used to target and dose mitigation measures to each of the sub-catchments in the Odder and Saltø catchments. In this way, the mitigation measures could be located in areas where the effect of reducing N emissions and meeting future targets is highest. This can help to obtain a better water quality in Danish streams and coastal waters.

## 5. Conclusions

Although the average N leaching for the two monitoring years was not very different in the three study catchments of Saltø (48 kg N/ha), Odder (58 kg N/ha), and Jegstrup (69 kg N/ha), very high differences were found in the average annual TN emissions to surface water, amounting to 30 kg N/ha, 40 kg N/ha, and 4.4 kg N/ha, respectively. The synchronous measurement within each of the three studied catchments revealed even higher variations in the TN emissions to surface water within sub-catchments (2 to 9 kg N/ha in Jegstrup catchment, 28 to 59 kg N/ha in Odder catchment, and 17 to 50 kg N/ha in Saltø catchment).

Our synchronous stream measurements proved to be useful when creating N emission and N retention maps at a local scale down to 200 to 1200 ha.

In the Jegstrup catchment, the N retention was above 80% in all three sub-catchments independent of using TN or $NO_3$-N. However, in the five Odder sub-catchments, there was a large variation (−18 to 53% using TN and 38 to 62% using $NO_3$-N), which was also the case for the seven sub-catchments in the Saltø catchment (−1 to 60% using TN, and 29 to 69% using $NO_3$-N). The average N retention based on TN and $NO_3$-N for the two monitoring years amounted to 93% and 97% in the Jegstrup catchment, 26% and 44% in the Odder catchment, and 44% and 57% in the Saltø catchment.

Our results clearly demonstrate that, when dealing with calculations of N retention in groundwater from $NO_3$-N leaching on agricultural fields using synchronous measurements at this finer resolution scale of 100–300 ha, the established balance between N leaching and resulting diffuse catchment N emissions should be based on $NO_3$-N. Moreover, it would

be difficult to achieve a finer scale than the one applied in this study, simply due to the uncertainty in mapping catchment boundaries, due to groundwater flows.

**Author Contributions:** Conceptualization, S.G.M.v.V. and B.K.; methodology, S.G.M.v.V., J.R., J.R.L., G.B.-M. and B.K.; software, S.G.M.v.V.; validation, S.G.M.v.V. and B.K.; formal analysis, S.G.M.v.V. and B.K.; investigation, S.G.M.v.V.; resources, S.G.M.v.V.; data curation, S.G.M.v.V., J.R. and J.R.L.; writing—original draft preparation, S.G.M.v.V.; writing—review and editing, S.G.M.v.V., J.R., J.R.L., G.B.-M. and B.K.; visualization, S.G.M.v.V.; supervision, J.R.L. and B.K.; project administration, S.G.M.v.V. and B.K.; funding acquisition, S.G.M.v.V. and B.K. All authors have read and agreed to the published version of the manuscript.

**Funding:** This research was part of the industrial Ph.D. project SENTEM funded by Innovation Fund Denmark grant 0153-00078B.

**Institutional Review Board Statement:** Not applicable.

**Informed Consent Statement:** Not applicable.

**Data Availability Statement:** Not applicable.

**Acknowledgments:** We would like to thank the technicians Uffe Mensberg, Henrik Stenholt, Dorte Nedergaard, and Marlene Skjærbæk for their enormous effort on collecting data in the field and analyzing data in the lab. Thanks to Ane Kjeldgaard for ArcGIS help and Henrik Tornbjerg and Jørgen Windolf for providing retention data on all sub-catchments from the Danish National Retention Map and scientific discussions. Thanks to Tinna Christensen for help with figure illustrations. Thanks to all farmers providing us with knowledge about the catchments and all partners in the EMIS GUDP project for their collaboration, especially Søren K. Hvid, SEGES INNOVATION, as project leader.

**Conflicts of Interest:** The authors declare no conflict of interest. The funders played no role in the design of the study, in the collection, analyses, and interpretation of data, in the writing of the manuscript, or in the decision to publish the results.

## Appendix A

**Table A1.** Q-Q relationship for all the synchronous stations in the Jegstrup, Odder, and Saltø catchments. This relationship was used to calculate the daily mean discharge at the synchronous stations in the two monitoring years.

| Jegstrup catchment stations | 2 | 3 | 4 | 5 | 6 | 7 |
| --- | --- | --- | --- | --- | --- | --- |
| Equation | y = 0.173x | y = 0.0662x + 0.3399 | y = 0.0432x + 2.3957 | y = 0.5912x + 7.7666 | y = 0.1955x + 14.674 | y = 0.2647x |
| Explanation value ($R^2$) | 0.9705 | 0.9843 | 0.7812 | 0.9962 | 0.9406 | 0.9566 |

| Odder catchment stations | 2 | 3 | 4 | 5 | | |
| --- | --- | --- | --- | --- | --- | --- |
| Equation | y = 0.7478x + 6.4106 | y = 0.0975x − 3.1895 | y = 0.3162x − 12.933 | y = 0.0613x − 1.1226 | | |
| Explanation value ($R^2$) | 0.987 | 0.9265 | 0.9749 | 0.8978 | | |

| Jegstrup catchment stations | 2 | 3 | | | | |
| --- | --- | --- | --- | --- | --- | --- |
| Equation | y = 0.1637x + 12.828 | y = 0.524x + 16.49 | | | | |
| Explanation value ($R^2$) | 0.7178 | 0.878 | | | | |

## Appendix B

Variations in discharge and TN concentrations measured at the synchronous stations in the sub-catchments and at the main gauging stations of Jegstrup catchment, Odder catchment, and Saltø catchment from 2014 to 2016. For the TN concentrations, the linear interpolation was used to obtain daily concentrations.

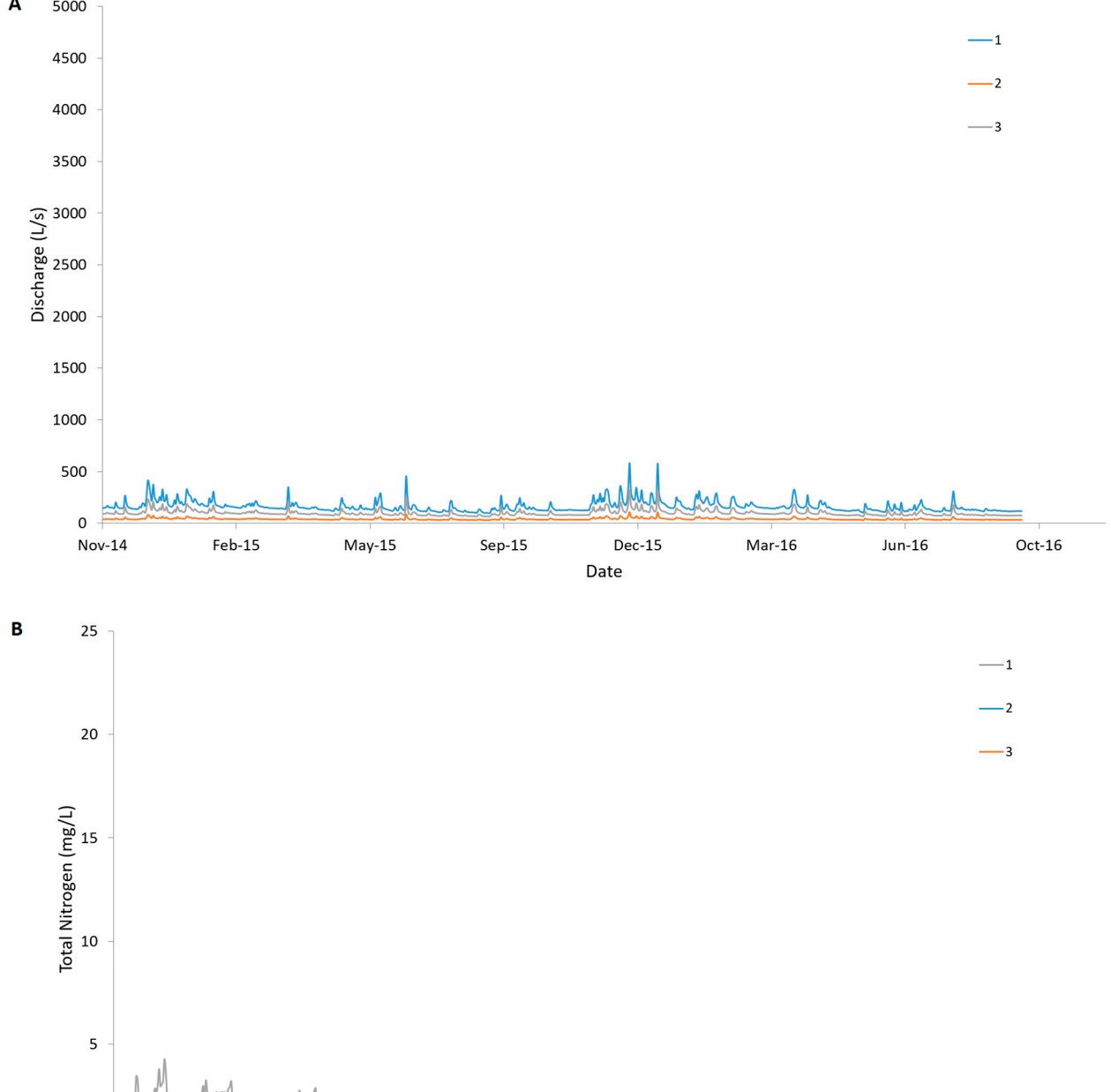

**Figure A1.** (**A**) Discharge measurement at the synchronous stations in the sub-catchment and at the main gauging station of the Jegstrup catchment in the period 2014 to 2016. (**B**) TN concentrations measured at the synchronous stations in the sub-catchment ant at the main gauging station of Jegstrup catchment in the period 2014 to 2016. Linear interpolation was used between weekly and fortnightly grab sample measurement to obtain daily measurements at the synchronous stations.

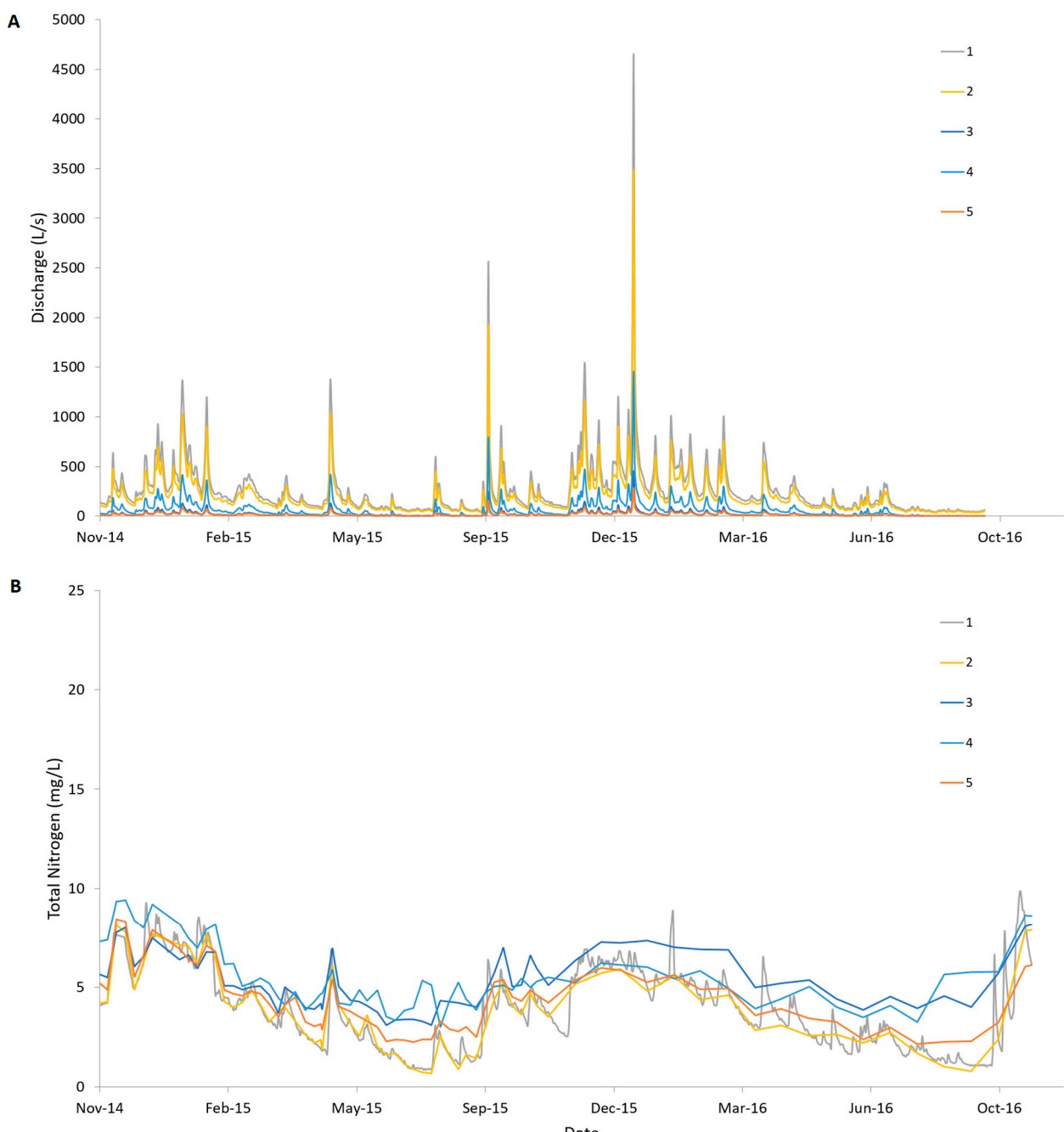

**Figure A2.** (**A**) Discharge measurement at the synchronous stations in the sub-catchments and at the main gauging station of Odder catchment in the period 2014 to 2016. (**B**) TN concentrations measured at the synchronous stations in the sub-catchment and at the main gauging station of Jegstrup catchment in the period 2014 to 2016. Linear interpolation was used between the weekly and fortnightly grab sample measurements to obtain daily measurements at the synchronous stations.

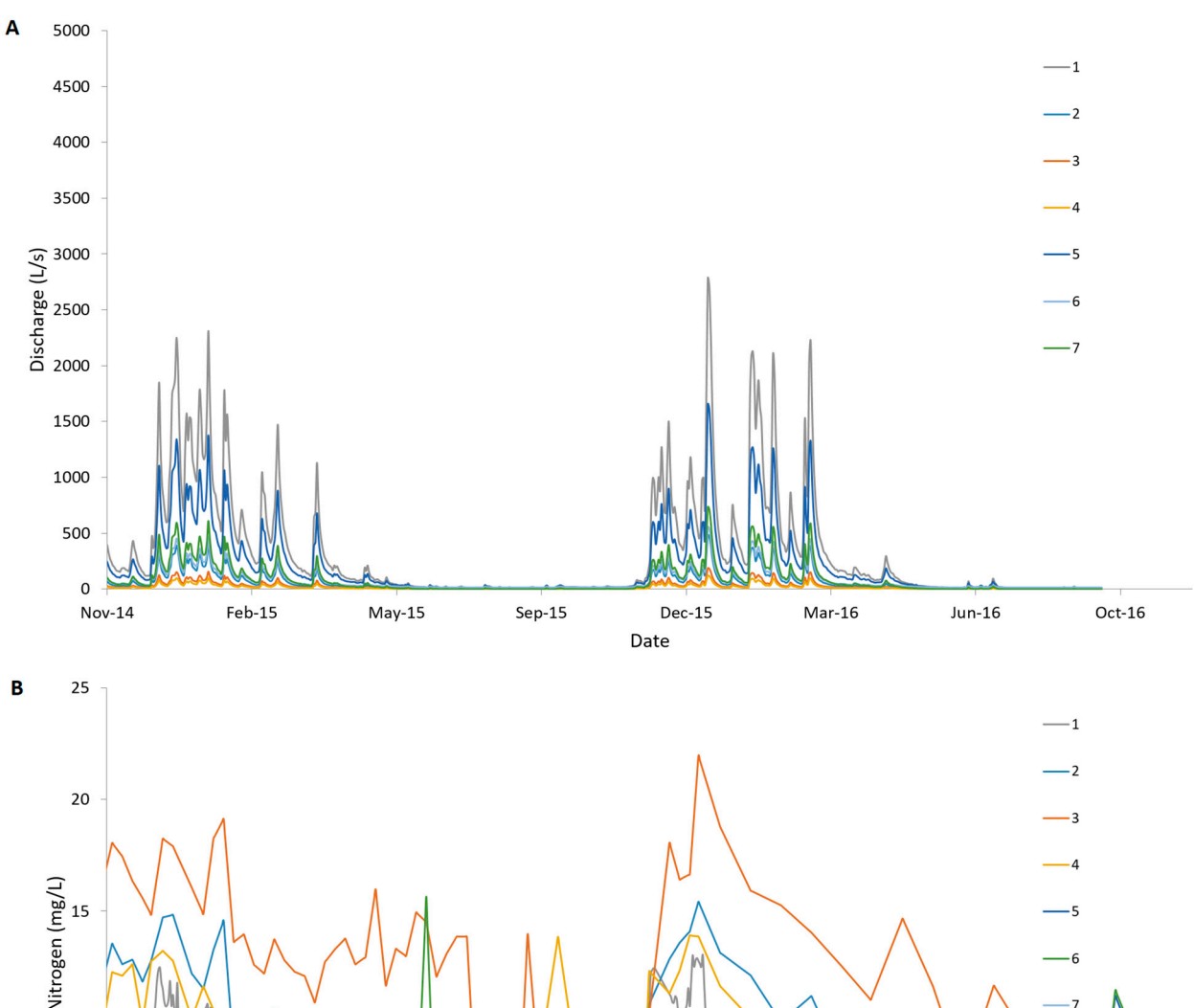

**Figure A3. (A)** Discharge measurements at the synchronous stations in the sub-catchments and at the main gauging station of Saltø catchment in the period 2014 to 2016. **(B)** TN concentrations measured at the synchronous stations in the sub-catchment and at the main gauging station of Jegstrup catchment in the period 2014 to 2016. Linear interpolation was used between weekly and fortnightly grab sample measurements to obtain daily measurements at the synchronous stations.

**Appendix C**

The delimitation of the main Jegstrup, Odder, and Saltø catchments into sub-catchments and the principle of the mass balance calculation used for estimation of the N transport and runoff for the individual sub-catchments (Table A2 and Figure A4).

**Table A2.** Table showing the delimitation of the main Jegstrup, Odder, and Saltø catchments into sub-catchments and the principle of the mass balance calculation when calculating the runoff and N transport for the individual sub-catchments. * By Head Water Catchment indication we mean that this sub-catchment was not affected by upstream sub-catchments. ** In the calculation, only the measurements at the main and synchronous stations were used for estimation of the runoff and N transport.

| Main Catchment | Catchment ID | Head Water Catchment * | Calculation Used for Measuring the N Transport and Runoff for the Individual Sub-Catchment ** |
|---|---|---|---|
| Jegstrup stream | 1 | | 1 = 1−3 |
| Jegstrup stream | 2 | X | |
| Jegstrup stream | 3 | | 3 = 3−2 |
| Odder stream | 1 | | 1 = 1−2−3 |
| Odder stream | 2 | | 2 = 2−4−5 |
| Odder stream | 3 | X | |
| Odder stream | 4 | X | |
| Odder stream | 5 | X | |
| Saltø stream | 1 | | 1 = 1−2−4−5 |
| Saltø stream | 2 | | 2 = 2−3 |
| Saltø stream | 3 | X | |
| Saltø stream | 4 | X | |
| Saltø stream | 5 | | 5 = 5−6−7 |
| Saltø stream | 6 | X | |
| Saltø stream | 7 | X | |

**Principle of mass balance calculation of the N transport for the individual sub-catchments**

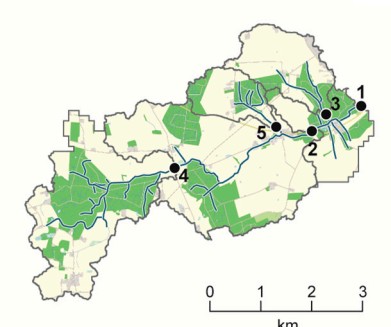

Ex. N transport for headwater sub-catchment 4 = N transport measured in sub-catchment 4

Ex. N transport for sub-catchment 1 = sub-catchment 1 - sub-catchment 2 - headwater sub-catchment 3

**ODDER STREAM, FILLERUP**

**Figure A4.** Conceptual figure showing the principle of the mass balance calculation when estimating the runoff and N transport for the individual sub-catchments (1 to 5) in the Odder catchment.

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
