# Peer review of "High Spatial Resolution Nitrogen Emission and Retention Maps of Three Danish Catchments Using Synchronous Measurements in Streams"

_water, doi:10.3390/w15030498_

Round 1
Reviewer 1 Report
This is a good work, clearly presented and of definite interest for many of the journal readers. I have only a few comments that the author may consider in their revision.
- I was rather confused by the term “groundwater N retention”, which is often used but in my view can generate some problems. From its calculation, described in section 2.5, I think it is the soil water N retention, not groundwater (which is conventionally interpreted as the water in saturated porous media, e.g. aquifers). Please clarify.
- Some level of detail in not quite necessary (e.g. section 2.2. can be shortened, as well as appendix A). Instead other important info are missing, e.g. more detail on the NLES4 model, which has a strong impact on the N balance and results.
- Also, some more information, even not much detailed, on groundwater would help a lot (e.g. size of groundwater catchment, depth to water table, connection with river network, etc). It might also provide some explanation regarding the negative TN retention that has been found.
- Regarding samples: was there any strategy for selecting locations? Any spatial analysis (e.g. geostatistical)? This has consequence on the data interpolation, e.g. linear.
- As far as I understand the results and the maps significantly rely on the leaching model: what is likely its impact? Some assessment or comment on uncertainty?
- The conclusions are rather short compared to the level of discussion brought in the paper. Perhaps some more take-home messages, even in bullets, would project this work beyond the particular case study adopted.
Reviewer 2 Report
The highest merit of article are measurements but reader can not see them.
You indicated that monitoring stations were installed in the streams at the outlet of each sub catchment (lines 168-169) but it is not clear for subcathment 1 (Saltø stream) where flow and N measured. If measurements were in the main river, then they describe whole watershed. How did you estimate flow and nitrogen for sub watershed 1? The same question for all watersheds.
You use models for estimation of nitrogen leaching and retention but description of models in Danish language (see your references). You have to explain models in more details if reader can not see model in English.
